# The Role of the Dysregulation of Long Non-Coding and Circular RNA Expression in Medulloblastoma: A Systematic Review

**DOI:** 10.3390/cancers15194686

**Published:** 2023-09-22

**Authors:** Ivan Martinez de Estibariz, Anastasija Jakjimovska, Unai Illarregi, Idoia Martin-Guerrero, Angela Gutiérrez-Camino, Elixabet Lopez-Lopez, Nerea Bilbao-Aldaiturriaga

**Affiliations:** 1Department of Genetics, Physical Anthropology and Animal Physiology, Faculty of Science and Technology, University of the Basque Country (UPV/EHU), 48940 Leioa, Spain; ivanmatute1711@gmail.com (I.M.d.E.); unai.illarregi@ehu.eus (U.I.); idoia.marting@ehu.eus (I.M.-G.); 2Department of Biochemistry and Molecular Biology, Faculty of Science and Technology, University of the Basque Country (UPV/EHU), 48940 Leioa, Spain; a.jakimovska7@gmail.com; 3Pediatric Oncology Group, Biocruces Bizkaia Health Research Institute, 48903 Barakaldo, Spain; angela.gutierrez@ehu.es

**Keywords:** medulloblastoma, lncRNA, circRNA, expression, miRNA

## Abstract

**Simple Summary:**

The role of long non-coding and circular RNAs in medulloblastoma, the most prevalent cerebral childhood cancer, has been widely studied from distinct perspectives, such as for the identification of potential biomarkers for diagnosis or prognosis. Moreover, their precise involvement in the origin and progression of the disease was one of the milestone discoveries regarding these molecules. New multi-omic techniques have enabled researchers to advance in the knowledge of medulloblastoma, adding complexity to the matter. Therefore, it is necessary to perform an update on all the evidence in the study of non-coding RNAs in this pathology. In this systematic review, we aimed to extract conclusions regarding the role of long non-coding and circular RNAs as biomarkers in medulloblastoma through the revision of the literature about their expression patterns and functional implications.

**Abstract:**

Medulloblastoma (MB) is the most common malignant brain tumor in childhood. Although recent multi-omic studies have led to advances in MB classification, there is still room for improvement with regard to treatment response and survival. Therefore, identification of new and less invasive biomarkers is needed to refine the diagnostic process and to develop more personalized treatment strategies. In this context, non-coding RNAs (ncRNAs) could be useful biomarkers for MB. In this article, we reviewed the role of two types of ncRNAs, long non-coding (lncRNAs) and circular RNAs (circRNAs), as biomarkers for the diagnosis, subgroup classification, and prognosis of MB. We also reviewed potential candidates with specific functions and mechanisms of action in the disease. We performed a search in PubMed and Scopus using the terms (“long non coding RNAs” OR ”lncRNAs”) and (“circular RNAs” OR ”circRNAs”) AND ”medulloblastoma” to identify biomarker discovery or functional studies evaluating the effects of these ncRNAs in MB. A total of 26 articles met the inclusion criteria. Among the lncRNAs, the tumorigenic effects of the upregulated *lnc-IRX3-80* and *lnc-LRRC47-78* were the most studied in MB. Among the circRNAs, the upregulation of *circSKA3* and its functional impact in MB cell lines were the most consistent results, so this circRNA could be considered a potential biomarker in MB. Additional validation is required for many deregulated lncRNAs and circRNAs; therefore, further studies are warranted.

## 1. Introduction

Medulloblastoma (MB) is an aggressive brain tumor accounting for 30% of all malignant central nervous system (CNS) cancers in childhood [1]. MB usually originates in the cerebellum or posterior fossa and can spread through the brain and spine via the cerebrospinal fluid (CSF) [2]. Most cases occur within the first nine years of life and are more common in males than in females [3]. Regarding its clinical features, molecular characteristics, and survival rates, MB is a very heterogeneous disease. Multi-omic studies, including whole exome sequencing (WES), transcriptomics (RNA-seq), and methylation, have allowed the classification of MB into four different molecular subgroups for risk classification, leading to an improvement in the overall survival of MB patients due to different treatment approaches [4]. According to the latest World Health Organization (WHO) classification in 2021, MBs can be classified as WNT MB, Sonic hedgehog (SHH) MB (*TP53* mutated or *TP53* wild-type), and non-WNT/SHH Group 3 (Gr3) and Group 4 (Gr4) MBs [2,5].

The WNT MB subgroup accounts for ~10% of all MBs and usually has an excellent prognosis, being classified as a low-risk (LR) MB. This subgroup commonly exhibits upregulation of WNT-responsive genes, driving cell growth and proliferation [6]. SHH MBs account for ~30% of all MBs and usually exhibit upregulation and mutations of SHH-pathway-related genes. These patients are classified into *TP53* wild type and *TP53* mutated groups, with remarkable differences in prognosis (75% and 40% survival rates, respectively), the former being classified as intermediate-risk (IR) and the latter as high-risk (HR) MB [7]. Finally, Gr3 and Gr4 MBs are the most common subgroups, accounting for ~25% and ~35% of all MBs, respectively. These subgroups, considered the most aggressive, are classified into the HR group [8]. The most differential feature of these MB subgroups is *MYC* amplification and the gain or loss of chromosome 17 q. Additionally, Gr3 MBs show an exclusive upregulation of *GFI1*/*GFI1B* oncogenes, which are transcriptional repressors that regulate cell fate decisions in the hematopoietic system, establishing this subgroup as the most aggressive of all the subgroups [2,9].

Even though advances in the molecular MB classification have improved patient risk stratification, one-third of patients are still classified as HR, and 20–25% of patients classified as LR or IR are incurable. Therefore, additional biomarkers are still needed to improve the treatment strategies.

It is worth noting that all the tumor biomarkers used for classification have been based on studies of the protein-coding landscape, setting aside the non-coding regions of the genome. In this context, the relevance of the predominant portion of the genome, referred to as non-coding RNAs (ncRNAs), is only now coming into consideration for the identification of new biomarkers to improve the MB molecular profiling.

Based on their length, ncRNAs can be divided into long ncRNAs and short ncRNAs, depending on whether they are longer or shorter than 200 nucleotides [10]. Among the long ncRNAs, lncRNAs, a specific ncRNA type, have great potential because they are altered in many cancer types due to their regulatory roles in transcription, translation, and miRNA function, as further described in Figure 1. Interestingly, 40% of human lncRNAs are expressed strictly in the brain, therefore contributing to the onset of tumorigenesis of brain tumors like MB [11]. Indeed, single lncRNAs for the molecular identification of pediatric MB subgroups have been proposed, although these require functional validation studies [12].

More recently, a new type of ncRNA, termed circular RNA (circRNA), has emerged as a promising biomarker in brain cancer [11]. CircRNAs are covalently closed-loop RNA molecules derived from 5′ to 3′ back splicing junctions that are generally transcribed from protein-coding genes known as parental genes and formed by 2–3 exons [13]. One of the advantages of these RNA molecules is that, given their circularity, they are much more stable molecules than linear RNAs, and, due to this stability, they are highly expressed in the brain [14]. Currently, the main hypothesis of action involving circRNAs and cancer development is that these RNA molecules act as miRNA sponges, thus regulating the expression of different miRNA target genes. As a result, they affect mechanisms such as angiogenesis, neuronal plasticity, autophagy, apoptosis, and inflammation, and their deregulation could contribute to cancer pathogenesis [15]. A study performed by Satoh et al. identified the first miRNA sponge in this tissue, *ciRS-7*, for *miR-7*, highlighting the potential of these molecules in MB disease [16]. In recent years, RNA-binding proteins (RBP) have been proposed as crucial factors involved in the mechanism of action of circRNAs. These proteins are widely involved in several regulatory processes, such as RNA shearing, translocation, sequence editing, intracellular localization, and translational control [17]. As depicted in Figure 1, circRNAs can recruit RBPs and thus contribute to their function.
Figure 1Functions and mechanisms of action of lncRNAs and circRNAs. (**A**). CircRNAs promote and/or inhibit parental gene expression [18]. Regulation of transcription: guiding of chromatin-modifying enzymes and alteration of DNA conformation (guide lncRNAs); removal of regulatory factors bound to DNA (decoy lncRNAs); promotion of ribonucleoprotein complex formation (scaffold lncRNAs). (**B**). CircRNAs serve as primary miRNA precursors, which are later processed into mature miRNAs (**C**). CircRNAs recruit proteins to build complexes. (**D**). CircRNAs enhance the function of protein complexes (**E**). LncRNAs promote the regulation of mRNA translation. (**F**). Some lncRNAs and circRNAs can be translated into active small peptides. (**G**). CircRNAs participate in protein scaffolding. (**H**). LncRNAs and circRNAs can be endogenous competitors acting as miRNA or RBP sponges. Abbreviations—circRNA: circular RNAs; lncRNA: long non-coding RNA; miRNA: micro-RNA; RBP: RNA-binding protein; ORF: open reading frame. Figure adapted from Illarregi U. et al., 2022 [19] and Kristensen L.S. et al., 2018 [20], CC BY 4.0 (https://creativecommons.org/licenses/by/4.0/).
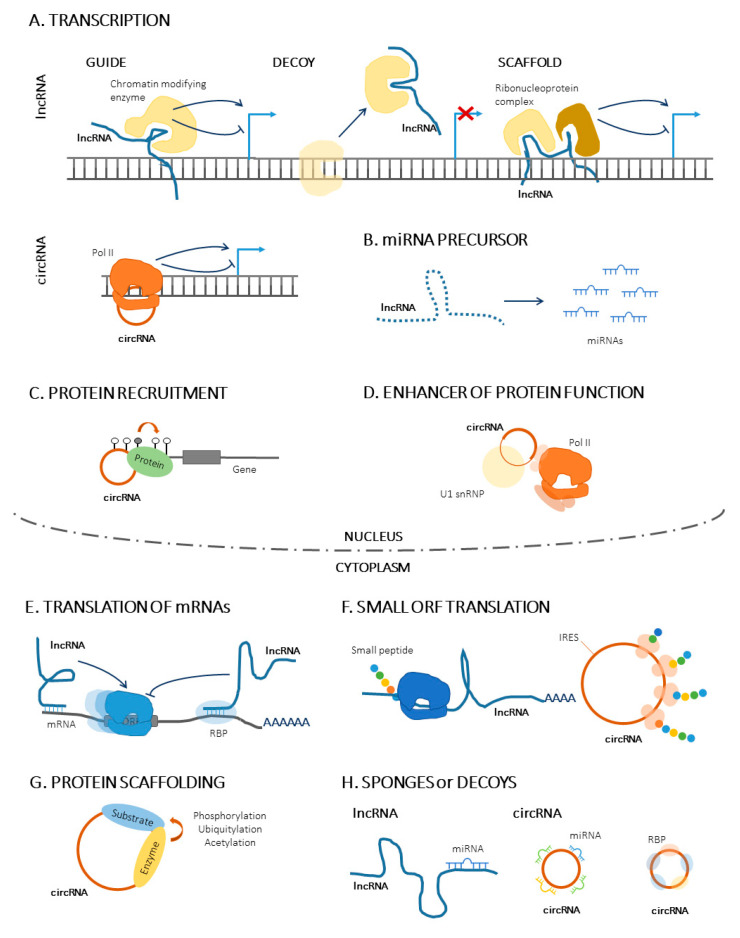



As the study of lncRNAs and circRNAs is an emerging field, it is necessary to perform an exhaustive study of all the literature focused on analyzing the role of these potential biomarkers in MB. Therefore, herein, we perform a systematic review of the expression profiles of lncRNAs and circRNAs as biomarkers for diagnosis, subgroup characterization, and prognosis in patients with MB, as well as of their functional characterization.

## 2. Materials and Methods

### 2.1. Search Strategy

Two exhaustive searches were performed in both the PubMed (https://www.ncbi.nlm.nih.gov/pubmed; accessed on 25 September 2022) and Scopus (https://www.scopus.com/home.uri; accessed 30 September 2022) bibliographic databases [21] to identify studies that examined lncRNA and circRNA expression or function in MB with regard to diagnosis, subgroup classification, prognosis, and treatment response. For the first search, we used the keywords (“long non coding RNAs“ OR “lncRNAs“) AND “medulloblastoma“ without data restrictions. Accordingly, for the second search, we used the keywords (“circular RNAs“ OR “circRNAs“) AND “medulloblastoma“ without data restrictions. The searches were last updated in February 2023. All references cited in the included studies were also reviewed to identify additional publications. Duplicated articles were removed using Jabref Software Version 5.1. (https://www.jabref.org; accessed 30 September 2022) (MIT License Copyright^©2023^–2022).

### 2.2. Inclusion and Exclusion Criteria

The inclusion criteria permitted the selection of original studies performed in MB patients or cell lines analyzing lncRNA or circRNA differential expression (DE) in relation to diagnosis, classification, prognosis, and treatment response or their function. Articles were excluded if they analyzed a disease other than MB, non-coding RNAs other than lncRNAs or circRNAs, features other than RNA expression, or organisms other than humans. In addition, articles not published in English were also excluded. I.M.d.E. and A.J. independently screened the studies and collected the relevant information. Any discrepancies concerning eligibility were resolved through discussion.

### 2.3. Data Extraction

For each study that met the criteria for inclusion, the following information was collected: the number of patients and controls involved, type of material used (such as cerebellum tissue samples, MB cell lines, etc.), classification by subgroup, method used, and parameters analyzed. Articles were classified into two main categories—biomarker discovery articles and functional analysis articles—depending on the study design. The biomarker discovery articles were defined as those studies in which DE analysis of a lncRNA and/or circRNA was performed in patient samples according to phenotype. In contrast, the functional analysis category included studies focused on describing the mechanism of action of the differentially expressed ncRNAs.

The discovery articles were evaluated based on a modified version of a protocol from the Oxford Centre for Evidence-based Medicine, which assesses the quality of individual studies (shorturl.at/gBOR8). The ratings were as follows: (i) Randomized clinical trials or systematic reviews with meta-analysis: 1; (ii) Prospective comparative cohort trials or well-designed controlled trials without randomization: 2; (iii) Case-control studies or retrospective cohort studies: 3; (iv) Case series with or without intervention or cross-sectional studies: 4; (v) Expert opinions and case reports: 5. The functional analysis studies were evaluated based on an in-house scale of the criteria. Each criterion was assigned a different number of points to reflect its relative relevance in assessing the comprehensive nature of functional analysis studies. The scale consists of 8 different aspects with specifically assigned values: the study uses MB cells (1 pt); sex distribution reflected that of the disease (1 pt); the use of two or more different cell lines (2 pts); the analysis of at least two cancer phenotypes (1 pt); the employment of either xenografts or cell lines derived from patients (3 pts); the inclusion of rescue experiments (2 pts); the investigation of a mechanism of action for the DE ncRNA (1 pt); the design includes data from patients (1 pt). After assigning the corresponding points to each article, the articles were ranked as follows: Q1: articles with 11–12 pts; Q2: articles with 9–10 pts; Q3: articles with 6–8 pts; Q4: articles with 4–5 pts; Q5: articles with 3 pts or less. The inclusion criterion was limited to articles falling within the Q1, Q2, or Q3 categories. By encompassing various aspects such as cell line diversity, clinical relevance, and mechanistic insight, we aimed to picture the quality of the functional evidence in lncRNAs and circRNA studies.

In this review, we included all the lncRNAs and circRNAs that were studied using the quantitative reverse transcription polymerase chain reaction (RT-qPCR). If sequencing or array expression was performed, we used all the information regarding the lncRNAs or circRNAs recorded in the article or Appendix A. LncRNA nomenclature was standardized using Ensembl codes from Ensembl GRCh38.p13 (https://www.ensembl.org/index.html; accessed 10 October 2022 ) and LNCipedia version 5.2 (https://lncipedia.org/; accessed 10 October 2022 ) when available. To standardize the circRNA nomenclature, we used Ensembl GRCh38.p13 (https://www.ensembl.org/index.html; accessed 5 October 2022) and circBase database (http://www.circbase.org/; accessed 5 October 2022) [22]. This systematic review was conducted following the PRISMA (Preferred Reporting Items for Systematic Reviews and Meta-Analyses) guidelines and has not been registered [23] (Appendix A).

## 3. Results

### 3.1. Systematic Search Results

#### 3.1.1. LncRNAs

A total of 106 entries were found in the PubMed and Scopus searches (Figure 2). Duplicates (*n* = 29) were removed, and 51 entries were eliminated after abstract screening because they did not meet the criteria for inclusion.

After full-text analysis of the remaining 38 articles, 8 others were excluded because they did not include data on lncRNAs, studied a disease other than MB, or used only mouse models. Finally, 18 articles were included in the review, of which 3 performed lncRNA expression discovery analysis in MB patients, 13 performed functional analysis of lncRNAs, and 2 performed both discovery and functional studies. After reviewing the references of the included studies, no other studies were added.

#### 3.1.2. CircRNAs

A total of 45 entries were initially identified (Figure 2). Of these, 15 duplicates were removed. Of the remaining entries, 18 were discarded after reading their abstract because they clearly did not meet the inclusion criteria.

The remaining 12 full-text articles were reviewed in detail to exclude those that did not meet the required criteria. After detailed review, four articles were excluded because they did not analyze circRNAs, did not focus on MB, or did not provide available full-text versions. Finally, eight articles were included that examined the role of circRNAs as biomarkers in human MB. All of them were discovery studies analyzing circRNAs as potential diagnosis biomarkers (*n* = 7) or subgroup biomarkers (*n* = 1). Four of them were also functional studies that analyzed the effect of a specific circRNA in different cancer phenotypes in human MB cell lines. No additional study was included after reviewing the references of the included studies.

### 3.2. Biomarker Discovery Studies in Medulloblastoma

#### 3.2.1. LncRNAs

All five articles were assessed as Quality 3 (Q3) articles according to the established guidelines.

In the studies that investigated the expression profile of MB patients, a large number of deregulated lncRNAs were detected, indicating their significance in the pathogenesis and prognosis of MB [12,24,25,26,27]. However, due to the different designs of the studies, there was no overlap in the lncRNA signatures that we could extract, and we considered each of the studies separately. Taking into account the sheer number of deregulated lncRNAs, analyzing all the lncRNAs might make it difficult to identify the key players. Therefore, we listed the top five deregulated lncRNAs within each subgroup from each study in Table 1. By narrowing the selection to the top five, we can provide a more focused and meaningful overview of the most pronounced changes in expression.

First, one study compared the expression profiles of 41 MB patients from all four subgroups with 5 normal cerebellum tissues to investigate the expression of six lncRNAs that had been previously associated in other brain tumors with metastasis and disease aggressiveness [24]. This study found considerably higher levels of four of them, namely *MALAT1*, *lnc-BMP6-106* (in the paper referred as *HULC*), *NEAT1*, and *SNHG16*, in MB compared with healthy individuals (Table 1).

Secondly, three studies examined the expression profiles of lncRNAs in different MB subgroups [12,25,26]. In WNT MB patients, 199 deregulated lncRNAs were identified in comparison with healthy individuals [12]. In SHH MB, patient samples presented 145 deregulated lncRNAs in comparison with healthy individuals [12]. When comparing SHH patients with non-SHH patients (Gr3 and Gr4 patients), 328 deregulated lncRNAs were identified, with *OTX2-AS1* being the most negatively correlated lncRNA [25]. Concerning Gr3 MB compared with healthy individuals, 149 deregulated lncRNAs were identified [12]. Moreover, several lncRNAs were significantly overexpressed in Gr3 MB compared with other subgroups, with the top five being *lnc-STAP1-13*, *lnc-MYO3A-1*, *lnc-SLITRK1-1*, *lnc-AXIN1-1*, and *lnc-POU5F1B-5* [26]. Finally, 150 deregulated lncRNAs were found in Gr4 MB compared with normal cerebellum [12] (Table 1).

Many of the top five deregulated lncRNAs overlap across the different MB subgroups (Table 1). This expression pattern suggests that there could be shared drivers and regulatory mechanisms involving these common lncRNAs across all MB subgroups. Regarding overlapping MB biomarkers, *LINC00844* and *SOX2-OT* were downregulated in all MB subgroups compared with healthy cerebellum tissue. *OTX2-AS1* was upregulated in WNT, Gr3, and Gr4 MBs, while *DLEU2* was upregulated in SHH, Gr3, and Gr4 MBs. *Lnc-DLK1* (referred to in the paper as *MEG3*) was downregulated in both WNT and Gr3 MBs, while *lnc-SYT16-67* (referred to in the paper as *LINC00643*) was downregulated in WNT and Gr4 MBs. *BLACAT1* and *LINC01355* were both upregulated in Gr3 and Gr4 MBs. Apart from the overlapping lncRNAs found in Table 1, there were other deregulated lncRNAs overlapping between MB subgroups and functionally researched in MB cells, such as *lnc-IRX3-80*, which is overexpressed in WNT and SHH MBs, and *NEAT1*, which is upregulated in SHH and Gr3 MBs compared with healthy tissue [12]. Interestingly, *MIR100HG*, which is one of the top five downregulated lncRNAs in Gr3 MBs, was also found among the upregulated lncRNAs in Gr4 MB compared with normal cerebellum [12], showing a more subgroup specific expression. Additionally, it is worth noting that specific lncRNAs have been reported to exhibit deregulation in certain subgroups in more than one study. For instance, *lnc-LRRC47-78* [25,29,30] and *HHIP-AS1* [12,25] have been documented to be overexpressed in the SHH subgroup. Interestingly, an intriguing association was observed between *lnc-LRRC47-78*′s overexpression and unfavorable patient outcomes in the *TP53* wild-type subgroup [30].

Finally, another study analyzed transcriptomic profiles of 175 publicly available MB patients’ datasets from all four subgroups to map subgroup-specific lncRNAs [23]. The study found that an 11-lncRNA signature (*MIR100HG*, *lnc-CFAP100-4*, *ENSG00000279542*, *lnc-ABCE1-5*, *USP2-AS1*, *lnc-RPL124*, *OTX2-AS1*, *lnc-TB1D16-3*, *ENSG00000230393*, *ENSG00000260249*, and *lnc-CCL2*) could accurately classify MB subgroups with an average < 7% class error rate.

Concerning the potential of lncRNAs as prognosis biomarkers, several lncRNAs have been suggested as prognosis biomarkers for MB in the articles reviewed. In the comprehensive study utilizing the MAGIC microarray dataset, Joshi et al. employed Cox proportional hazards regression to analyze the expression levels of 621 lncRNAs, identifying 17 lncRNAs as potential prognostic markers [27]. Out of the 17 identified lncRNAs, 10 were linked with a positive prognosis (*lnc-TMEM258-3*, *lnc-ZNRF3-AS1*, *lnc-PRR34-1*, *lnc-KLF3-AS1*, *lnc-FOXD4L5-25*, *lnc-TTC28-AS1*, *lnc-TMEM121-3*, *LINC01152*, *lnc-MAP3K14-AS1*, and *AC209154.1*) and 6 with poor prognosis (*lnc-H19*, *lnc-RRM2-3*, *LINC01551*, *LINC00336*, *lnc-CDYL-1*, and *AL139393.2*) [27]. Additionally, Mutlu et al. identified a significant association between high expression levels of *MALAT1* and *SNGH16* and shorter overall survival in patients [24]. Finally, considering the different subgroups, upregulation of *HAND2-AS1* was linked to a poor prognosis in WNT MB [12] and upregulation of *DLEU2* was associated with poor prognosis in SHH MB patients, while both high expressions of *DLEU2* and *DSCR8* were linked to a poor prognosis in Gr3 MB [12] and low expression of *XIST* was linked to a poor prognosis in Gr4 MB patients [12].

#### 3.2.2. CircRNAs

Eight articles were included in this section, and all of them were evaluated as Quality 3 (Q3) articles according to the established guidelines.

Seven of these articles searched for circRNAs deregulated in MB patients vs. healthy controls to identify circRNAs that could be used as biomarkers for diagnosis in this disease (Appendix A). These studies permitted the identification of five circRNAs (*circSKA3*, *circSMARCA5*, *circLPAR1*, *circFAM13B*, and *circHIPK3*) that were significantly deregulated between MB patients and healthy controls in at least two different studies. *CircSKA3* was the only circRNA found to be deregulated in more than two different studies, being analyzed in four out of the seven studies. Interestingly, *circSKA3* was upregulated in all of them [31,32,33,34]. Moreover, *circSMARCA5*, *circLPAR1*, *circFAM13B*, and *circHIPK3* were analyzed in two different studies and were found to be downregulated in the MB group compared with healthy controls in both studies [35,36] (Table 2).

Finally, the last article included a discovery study that analyzed circRNAs as subgroup biomarkers in MB [38]. Four DE analyses were performed, including the four different subgroups of MB vs. healthy controls. In this context, different circRNAs appeared to be deregulated in each subgroup comparison, establishing unique circRNA signatures as biomarkers for subgroup classification in MB (Table 3). Interestingly, some circRNAs are commonly deregulated in three out of the four subgroups, such as *circSYNEI1*, which was downregulated in all subgroups except Gr4, or *circCDYL2*, *circMCU*, and *circPHF21A*, which were downregulated in all subgroups except WNT. Moreover, we can observe that *circKHDRBS2* is commonly upregulated in Gr3 and Gr4 MBs.

### 3.3. Functional Studies in Medulloblastoma

#### 3.3.1. LncRNAs

Out of the fifteen functional studies, four were evaluated as Q1 [29,39,40,41], six as Q2 [25,26,42,43,44,45], and five as Q3 [30,46,47,48,49].

The function of *lnc-IRX3-80* (referred to in the papers as *CRNDE*) [48,49] and *lnc-LRRC47-78* (referred to in the papers as *TP73-AS*) [29,30] in MB was each studied in two independent studies (Table 4). The remaining eleven lncRNAs were only studied in single studies, each of them focused on a specific lncRNA [25,26,39,40,41,42,43,44,45,46,47] (Appendix A). Eight of the thirteen lncRNAs studied (*lnc-IRX3-80*, *lnc-LRRC47-78*, *lnc-FAM84B-15* (referred to in the paper as *CCAT1*), *LOXL1-AS1*, *MIR100HG* (referred to in the paper as *LINC-NeD125*), *lnc-FGF1-9* (in the paper referred as *SPRIGHTLY*), *NEAT1*, and *HHIP-AS1*) were also found to be differentially expressed in one or more of the discovery studies.

The investigations corresponding to *lnc-IRX3-80* and *lnc-LRRC47-78* performed targeted knockdown of the respective lncRNAs to investigate their impact on tumorigenesis-associated phenotypes. Such interventions resulted in elevated rates of apoptosis and decreased cell proliferation [29,30,48]. In addition, expression levels of *lnc-IRX3-80* were significantly higher in cisplatin-exposed MB cell lines compared with non-treated MB cell lines, suggesting that *lnc-IRX3-80* may be associated with drug resistance in MB [49]. Notably, Sun et al. [49] reported a decrease in cellular invasion upon *lnc-IRX3-80* knockdown, while Li et al. [29] observed a similar outcome in response to *lnc-LRRC47-78* silencing. These studies also examined the interaction between the lncRNAs and miRNAs via targeted experiments. Specifically, *lnc-IRX3-80* downregulated *miR-29c-3p*, while *lnc*-*LRRC47-78* was observed to suppress *miR-494-3p*, implying their potential as miRNA sponges [29,49] (Table 4).

#### 3.3.2. CircRNAs

With respect to functional analyses, *circSKA3* was the only circRNA studied by a functional approach in MB, and it was analyzed in four different articles. In these studies, the role of *circSKA3* was studied in different cancer phenotypes through overexpression and/or silencing studies in MB cell lines. Two articles were evaluated as Q2 [31,34], while the remaining articles were evaluated as Q3 and Q1 [32,33] (Table 5).

As seen in Table 5, all the studies performing *circSKA3* knockdown through specific small interfering RNAs (siRNAs) obtained reduced proliferation, invasion, and migration rates in silenced MB cell lines. Conversely, apoptosis and cell cycle arrest rates were increased when *circSKA3* was silenced. Moreover, Liu X., et al. [24] and Zhao X., et al. [34] performed *circSKA3* overexpression analysis through overexpression plasmids. In this context, the different cancer phenotype patterns were completely reversed compared with *circSKA3* silencing. Finally, three of the studies included miRNA targeting experiments in which the interaction between *circSKA3* and miRNAs was analyzed [31,33,34]. Three different miRNAs (*miR-383-5p*, *miR-520*, and *miR-326*) appeared to be downregulated by *circSKA3*, confirming that this circRNA may act as miRNA sponge.

## 4. Discussion

The study of the non-coding region of the genome in medulloblastoma is gaining interest as it may explain the regulatory mechanisms of the physiopathology of the disease. This systematic review focused on the role of lncRNAs and circRNAs as biomarkers in MB and their potential functions. We reviewed biomarker discovery studies searching for deregulated lncRNAs and circRNAs in MB to define an ncRNA signature for diagnosis and subgroup classification. Moreover, the functional role of these ncRNAs in MB was reviewed to define those ncRNAs that have a key function in the development of MB.

First, lncRNAs were studied more thoroughly than circRNAs, with 15 results regarding the involvement of various lncRNAs in MB tumorigenesis [25,26,29,30,39,40,41,42,43,44,45,46,47,48,49]. The main difference with the circRNA studies was the analysis of more subgroup comparisons. Interestingly, the results of clustering MB patients with highly variable lncRNAs were consistent with molecular and clinical subgroups, suggesting that these genes are functionally relevant and may play a critical role in MB tumorigenesis [27]. With the available information, we were unable to identify subgroup-specific lncRNAs with certainty, but there were some lncRNAs that showed specific deregulation toward a certain subgroup in more than one work, such as *lnc-LRRC47-78* [25,29,30] and *HHIP-AS1* [12,25] overexpression in the SHH subgroup, and *MIRHG100*, which was upregulated in the Gr4 and downregulated in the Gr3 subgroups. In addition, the studies revealed an overlapping signature of deregulated lncRNAs across different MB subgroups, which may be important for a better understanding of the underlying biology of the disease and the possible development of targeted therapeutic approaches. Moreover, the functional effects of 13 lncRNAs were studied, but only *lnc-IRX3-80* (*CRNDE*) [48,49] and *lnc-LRRC47-78* (*TP73-AS1*) [29,30] were studied in two independent studies. Furthermore, although several prognostic biomarkers were mentioned in the studies, these markers have not yet undergone comprehensive investigation and validation to be defined as reliable prognostic indicators. The identification and characterization of prognostic biomarkers, along with an understanding of their underlying mechanisms of action, would significantly enhance our understanding of the factors contributing to the poor prognosis and aggressiveness observed in specific subgroups, such as Gr3 and Gr4.

Out of the lncRNAs mentioned in the results, we have chosen five particularly intriguing ones for further discussion, namely *NEAT1*, *MIR100HG*, *lnc-LRRC47-78*, *lnc-IRX3-80*, and *HHIP-AS1*. The selection of these specific five lncRNAs was based on their dysregulation in several discovery studies and their functional investigation in the functional studies, highlighting their potential significance in MB.

Concerning *NEAT1*, this lncRNA exhibits overexpression in MB patients compared with normal tissue [12,24] and, specifically, within the SHH or Gr3 subgroups [12,24]. The overexpression of *NEAT1* plays a crucial oncogenic role in other solid tumors, such as lung cancer and breast cancer [50]. This suggests that *NEAT1* also might function as a significant oncogene within MB. In this context, a study delved into the functional aspects of *NEAT1* in MB and found its role as sponge for *miR-23a-3p* [40], a miRNA recognized as a tumor suppressor that is frequently downregulated in various cancers [51,52,53]. Moreover, *NEAT1*/*miR-23a-3p* sponging has been validated in multiple cancers, reinforcing the oncogenic significance of this interaction [54,55,56,57]. The sponging of *miR-23a-3p* by *NEAT1* in MB leads to an upregulation of glutaminase (GLS), which results in increased glutamine metabolism [40]. Interestingly, this metabolic shift is correlated with cisplatin resistance in MB cells [40]. This association implies that this lncRNA might contribute to an unfavorable treatment response, aligning with observations in other tumor types [50]. Since *NEAT1′*s oncogenic role in other tumor types goes beyond miRNA sponging, also involving interactions with RBPs to form paraspeckles and to impact gene regulation and cancer progression [58], these interactions need to be investigated in the context of MB.

Moving to *MIR100HG*, three studies showed that this lncRNA was overexpressed in Gr4 MB patients compared with normal cerebellum tissues [12,27,47]. Additionally, Kesherwani et al. showed that *MIR100HG* was one of the top five downregulated lncRNAs in Gr3 MB [12]. This expression trend of *MIR100HG* underscores its potential to serve as a discriminatory marker, improving the distinction between Gr3 and Gr4 MB patients. In this context, it is worth noting that *MIR100HG* was one of the eleven lncRNAs in the proposed signatures for accurately classifying MB patients into subgroups [27], suggesting once again the importance of this ncRNA in MB patient stratification. Interestingly, *MIR100HG* is a dysregulated emerging ncRNA in multiple tumors, where it can have either an oncogenic or tumor-suppressive role [59]. *MIR100HG* was further analyzed in a functional study, in which it was shown to derepress the expression of the major driver genes in Gr4 MB—*CDK6*, *MYCN*, *SNCAIP*, and *KDM6A*—by acting as competing endogenous RNAs (ceRNAs) and binding to three miRNAs—*miR-19a-3p*, *miR-19b-3p*, and *mir-106a-5p* [47]. In agreement with the subgroup-specific expression observed in patients, *MIR100HG* overexpression in Gr3 MB cells resulted in the development of Gr4-specific molecular features as well as a reduction in proliferation, invasion, and migration, all of which resembled Gr4 behavior [47]. Taken together, the potential of utilizing *MIR100HG* as a marker for identifying Gr4 MB patients, coupled with its possible role in driving the tumorigenesis of this subgroup, sets the stage for future investigations.

*Lnc-LRRC47-78* (also known as *TP73-AS1*) was upregulated in MB patients compared with healthy individuals [29,30]. Additionally, *lnc-LRRC47-78* exhibited differential expression in SHH versus non-SHH MB patients [25]. Within this subgroup, specifically in the *TP53* wild-type subgroup, high expression of *lnc-LRRC47-78* was correlated with a poorer prognosis, highlighting its potential as a prognostic indicator [30]. Interestingly, in a particular subgroup of glioblastomas associated with a more favorable prognosis, the *lnc-LRRC47-78* promoter is hypermethylated, resulting in its downregulation [60]. In MB cells, the *lnc-LRRC47-78* gene was hypomethylated, leading to its upregulation, which elevated the proliferation, migration, and survival rates [30]. This indicates that changes in methylation are key influences on the activity of *lnc-LRRC47-78* in brain tumors, including MB. These changes, whether causing upregulation or downregulation, are most probably closely associated with patient outcomes, highlighting the potential of *lnc-LRRC47-78*’s methylation status as a critical indicator for both prognosis and potential therapeutic avenues. According to the functional studies performed on MB, *lnc-LRRC47-78* serves as an oncogene by sponging *miR-494-3p*, thus elevating the expression of *EIF5A2* [29]. Interestingly, the activity changes in *EIF5A* were linked to cancer development, particularly its higher expression in GBM patient samples compared with normal glia cells [61]. However, further research is warranted to better understand the mechanisms involved in the *lnc-LRRC47-78/miR-494-3p/EIF5A2* axis in MB for potential targeted therapeutic approaches.

Regarding *lnc-IRX3-80*, also known as *CRNDE*, this was found to be upregulated in MB patients compared with adjacent non-cancerous tissue. Additionally, it was overexpressed in WNT and SHH patients compared with other MB patients [27,48]. *Lnc-IRX3-80* is overexpressed in many brain tumors [62]. For instance, in glioma, this lncRNA expressed oncogenic behavior through the mTOR signaling pathway [62]. In MB, high expression of *lnc-IRX3-80* seems to promote tumor growth and is associated with cisplatin resistance [48,49]. In fact, transcript levels of lncRNA *lnc-IRX3-80* are significantly higher in cisplatin-treated tumor cells [49]. In this scenario, *lnc-IRX3-80* negatively regulates the expression of *miR-29c-3p*, a miRNA that acts as a tumor suppressor and sensitizes MB cells to cisplatin [49]. *Lnc-IRX3-80*’s role as a ceRNA in MB holds promise for targeted therapies. By manipulating its interactions with miRNAs, crucial tumor-related genes could be influenced. However, *lnc-IRX3-80*’s influence extends beyond ceRNA interactions. For instance, its positive connection with the epidermal growth factor receptor (EGFR) in high-grade oligodendroglioma suggests diverse roles [63]. To unlock its full potential, further research into *lnc-IRX3-80*’s multifaceted interactions in MB is essential, offering opportunities for new therapeutic approaches.

Finally, *HHIP-AS1* was overexpressed in SHH MB patients compared with non-SHH MB patients [25] or to non-cancerous tissue [12]. In fact, the overexpression of this lncRNA serves as a hallmark of SHH-driven tumors, such as MB and atypical teratoid tumors (ATRT) [25]. This is explained by the fact that *HHIP-AS1* is a downstream target of the SHH signaling pathway and plays a crucial role in mediating its proliferative effects [25]. According to functional studies in MB, the tumorigenic effect of this lncRNA could rely on its direct binding to the mRNA of cytoplasmic dynein 1 intermediate chain 2 (*DYNC1I2*), attenuating its degradation by *miR-425-5p* [25]. By maintaining proper levels of *DYNC1I2*, *HHIP-AS1* sustains spindle assembly and chromosome segregation during cell division, impacting cell progression [25]. *HHIP-AS1’s* selective overexpression within the SHH subgroup, as well as its regulatory role in promoting tumorigenesis, hold potential for better patient stratification and targeted therapeutic approaches. However, further research is essential to fully comprehend the regulatory *HHIP-AS1/DYNC1I2/miR425* axis and its significance in SHH-driven neurogenesis and tumorigenesis.

On the other hand, five circRNAs (*circSKA3*, *circSMARCA5*, *circLPAR1*, *circFAM13B,* and *circHIPK3*) were found to be significantly deregulated in MB patients compared with healthy controls in at least two different studies [35,36]. Among them, *circSKA3* was the most studied, being concordantly significantly upregulated in the four studies in which it was analyzed. Additionally, these studies revealed that silencing *circSKA3* reduced proliferation, invasion, and migration rates in MB [31,32,33,34], while apoptosis and cell cycle arrest rates were increased [31,33,34]. Accordingly, its overexpression showed the opposite pattern for all the cancer phenotypes [31,34]. Considering all these data, an oncogenic role for *circSKA3* in MB tumorigenesis is supported. Indeed, miRNA sponging has been shown as a relevant mechanism of action of this circRNA. Three miRNAs (*miR-520*, *miR*-383, and *miR-326*) were identified as direct targets of *circSKA3*, being downregulated when *circSKA3* expression is increased in MB cells [31,33,34]. Among them, *miR-520* is an oncomiR that negatively regulates the *CDK6* protein, an essential regulator in cell-cycle progression. This data suggests that *circSKA3* could facilitate MB progression by targeting *miR-520*, leading to the upregulation of *CDK6* expression [31,64]. Similarly, *miR-383* inhibition by *circSKA3* could facilitate MB development [31] through its target, *FOXM1*, which is a critical proliferation-associated transcription factor widely overexpressed during the cell cycle in most human cancers [65]. Finally, *miR-326* is an oncomiR that inhibits *ID3*, a gene that has been shown to enhance MB cell proliferation. Conversely, degradation of *ID3* gene increased MB cell apoptosis and impaired tumor cell migration [66,67,68]. These data suggest that the upregulation of *circSKA3* leads to lower *miR-520/miR-383/miR-326* expression, which may enhance MB development, as apoptosis, migration or proliferation rates might be affected. Even though this circRNA has an important role in MB development, further investigation is needed regarding its proposed mechanisms of action.

The second most interesting circRNA, *circSMARCA5*, was found to be significantly downregulated in MB cell lines and in CSF compared with healthy control samples, as reported by Azatyan et al. and Lee et al. [35,36]. The expression level of *circSMARCA5* is accordingly downregulated in different cancer types, such as liver cancer or glioblastoma [69]. Interestingly, this oncogenic circRNA could play a regulatory role in tumorigenesis by acting as a miRNA sponge [69], as its ability to be a sponge for *miR-17-3p/miR-181b-5p* and *miR-4295* has been reported in two different studies on hepatocellular carcinoma and gastric cancer [70,71]. These miRNAs regulate the expression of *PTEN*, which can negatively regulate *PI3K/AKT* signal transduction, thus inhibiting or promoting tumor progression [71]. In fact, different studies have already shown that *miR-17-3p* and *miR-181b-5p* are usually upregulated in different cancer types, such as in colon and breast cancer, respectively [72,73]. In this context, the downregulation of *circSMARCA5* in MB samples could hypothetically explain MB tumor progression, as lower expression of this circRNA would result in higher expression of the mentioned miRNAs. As a result, *PTEN* expression would be reduced, resulting in a constitutively active *PI3K/AKT* signaling pathway and a potential increase in tumor proliferation and invasion.

While miRNA sponging is the most studied mechanism of action of this circRNA, interactions with RBPs have been also described in different cancer types. For instance, in glioblastoma, tumorigenic RBP *SRSF1* was found to bind to *circSMARCA5* at multiple sites, with this circRNA functioning as a tumor suppressor, regulating the activity of *SRSF1/SRSF3/PTB* axis and preventing tumor angiogenesis and migration [74]. As for the hypothesis proposed for *circSMARCA5* and *miR-17-3p/miR-181b-5p*, the downregulation of *circSMARCA5* in MB may result in increased expression of the *SRSF1/SRSF3/PTB* axis, promoting MB development. However, further studies regarding these interactions in MB are needed to decipher the role of *circSMARCA5* in the disease.

In addition, *circLPAR1* is another circRNA that is downregulated in MB samples [35,36], and it is reported to be especially expressed in the brain [75,76]. This downregulation agrees with previously published observations in bladder cancer, in which the levels of *circLPAR1* were also decreased [77]. Similarly to *circSMARCA5*, *circLPAR1* may contribute to its role through miRNA targeting as it has been shown to target *miR-762*, a known oncomiR, in bladder cancer [75,77]. Because this circRNA is downregulated in MB, if the same mechanism applies, *miR-762* would be overexpressed and its target gene *IRF7* would be inhibited, as in a study published on breast cancer [77]. Furthermore, another study involving gastric cancer revealed that this miRNA was aberrantly upregulated in tumor tissue and cell lines. This deregulation increases *PI3K/AKT* signaling pathway activity and, thus, promotes cell proliferation [78]. In summary, the deregulation of this gene and potential miRNA targets could be a plausible mechanism of action concerning the downregulation of this circRNA in MB. However, further studies regarding this miRNA and other potential mechanisms like interactions with RBPs are needed in MB samples.

Of note, two studies found *circFAM13B* to be downregulated in MB [35,36]. In contrast, this circRNA was shown to be upregulated in hepatocellular carcinoma. In that tumor type, *circFAM13B* contributes to tumorigenesis through *miR-212* downregulation and activation of the *E2F5* gene, which inhibits the p53 signaling pathway [79]. As the pattern of deregulation is different, the mechanism of action of downregulation of *circFAM13B* in MB development could follow a different pathway and needs further confirmation. Interestingly, in a study published by Lv et al., *circFAM13B* was downregulated in bladder cancer [80]. In this context, a potentially tumor-driving interaction between *circFAM13B* and the RBP *IGF2BP1* was revealed. In bladder cancer, this RBP usually promotes tumor proliferation, migration, and invasion by regulating the expression of *MYC* and *FSCN1*. Considering this information, we propose the hypothesis that the downregulation of *circFAM13B* in MB may upregulate *IGF2BP1* expression levels and, thus, promote tumor growth through deregulation of the *MYC* and *FSCN1* oncogenes.

Finally, *circHIPK3* was downregulated in MB in the two studies in which it was analyzed [35,36]. *CircHIPK3* is a well-known circRNA involved in tumorigenesis. However, its expression is usually upregulated in different cancers [81]. In fact, two independent studies in glioma, another type of brain tumor [82,83], showed an overexpression of *circHIPK3* and its involvement in the disease by reducing the expression of *miR-421* and *miR-124*, which prevent cell proliferation and growth through *CDK6* inhibition [84]. Additionally, these miRNAs seem to be downregulated in different cancers, such as colorectal cancer, confirming the hypothesis that *circHIPK3* upregulation may reduce these miRNA levels [85]. Those miRNAs are commonly downregulated in MB, which does not fit with the proposed hypothesis that this is mediated by the upregulation of *circHIPK3*, as it has been shown in different cancers. Therefore, further studies are needed to clarify the role of this circRNA in MB.

Several limitations were faced while performing this systematic review. First, the included studies were very heterogeneous in their scope and methods. In addition, divergence in lncRNA and circRNA nomenclature or annotation used among articles made comparison among studies difficult. All of this led to a very limited identification of results that were validated in the various studies. Therefore, it was difficult to reach further conclusions. Furthermore, although we have used the two most extensive article databases and general keywords, it is possible that some additional articles have been missed by our search strategy.

Despite those limitations, this review gives some important evidence regarding lncRNA and circRNA deregulation patterns in MB samples, and the functional impact these molecules may have on MB development.

## 5. Conclusions

We can conclude that the presented data confirm that lncRNAs and circRNAs play an important role in MB. 

Many deregulated lncRNAs were identified in the four MB subgroups, and the observed heterogeneity of lncRNA expression in the different subgroups holds promise for a better stratification of patients. However, more comprehensive comparisons of lncRNA expression among the different MB subtypes subgroups should be conducted to elucidate their subgroup-specific dysregulation. Among the deregulated lncRNAs, few were functionally studied. Of note, *NEAT1*, *MIR100HG*, *lnc-LRRC47-78*, *lnc-IRX3-80*, and *HHIP-AS1* were more frequently found deregulated in MB or in specific subgroups. These might mediate their effects via the regulation of miRNA pathways, either by miRNA sponging or by binding to their targets. Performing further functional studies on the dysregulated lncRNAs would provide further evidence confirming their roles in MB, helping to identify new treatment targets and diagnostic tools. Given the considerable number of dysregulated lncRNAs and the various ways they can interact, extending beyond miRNA sponging, conducting more functional studies is essential to further validate and clarify their specific roles and involvement in MB.

Five circRNAs were identified as being significantly deregulated in at least two studies (*circSKA3*, *circSMARCA5*, *circLPAR1*, *circFAM13B*, and *circHIPK3*). *CircSKA3* contributes to MB development through the deregulation of *miR-520/CDK6* and *miR-383/FOXM1* pathway axes. The presented data also suggest that the downregulation of *circSMARCA5* and *circLPAR1* may affect MB pathogenesis through miRNA sponging, highlighting the interactions between *circSMARCA5/miR-17-3p/miR-181b-5p*/*PTEN* and *circLPAR1/miR-762/IRF7*. Therefore, these results represent an interesting view of the potential mechanisms of action involving three different deregulated circRNAs in MB pathogenesis. Nevertheless, considering the heterogeneity of subgroups of MB, studies on all the circRNAs are required, especially in statistically more powerful cohorts to establish reliable circRNA signatures. Additionally, studies analyzing these circRNAs and their targets in MB cell lines should be performed to confirm their mechanism of action.

Furthermore, we have shown that interactions between RBPs and these ncRNAs may play a crucial role in MB. Different mechanisms regarding *circSMARCA5-SRSF1* and *circFAM13B-IGF2BP1* interactions could be related to MB growth and development, as previously reported in various cancers. However, with the exception of miRNA sponging, the functional roles of lncRNAs and circRNAs in MB have hardly been investigated, as reported in this systematic review. This is particularly important for lncRNAs, as many different cancer-related functions have been described that may underly important mechanisms in the development of MB. This suggests that further studies are needed, particularly focusing on the different functions of these molecules in MB cells.

In summary, this systematic review supports the idea that MB diagnosis and prognosis could be greatly improved by identifying lncRNAs and circRNAs as biomarkers. However, limitations regarding small sample sizes with very few control samples and different methodological approaches imply the need for future studies to unveil the function of many lncRNAs as well as to discover and functionally explore circRNAs.

## Figures and Tables

**Figure 2 cancers-15-04686-f002:**
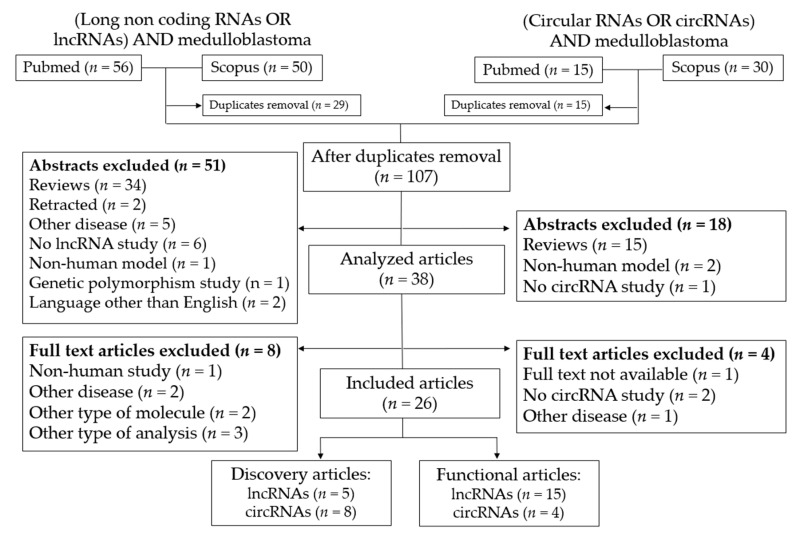
Flowchart of the systematic search. LncRNA- and circRNA-based systematic search results.

**Table 1 cancers-15-04686-t001:** Most significantly deregulated lncRNAs in MB from DE analysis studies.

Comparison	Expression	Studied Genes; DE	LncRNA Source	Method	Reference
41 MB vs. 5 normal cerebellum tissue		6 lncRNAs; 4 DE	Tissue RNA (RNeasy FFPE Mini Kit, Qiagen)	RT-qPCR (StepOnePlus)	[24]
	upregulated	*MALAT1*, *lnc-BMP6-106*, *NEAT1*, *SNHG16*			
8 WNT vs. 5 normal cerebellum tissue		NA; 199 DE	Publicly available datasets (GSE37418, GSE44971, GSE85217)	Array (Affymetrix U133 Plus2 array)	[12]
	upregulated	*EMX2OS*, *OTX2-AS1*, *PGM5-AS1*, *DSCR8*, *LOXL1-AS1*			
	downregulated	*LINC00461*, *lnc-DLK1*, *LINC00844*, *LINC00643*, *SOX2-OT*			
10 SHH vs.5 normal cerebellum tissue		NA; 145 DE			
	upregulated	*NEAT1*, *DLEU2*, *PRR34-AS1*, *LINC01355*, *MIRLET7BHG*			
	downregulated	*LINC00844*, *MIR124-2HG*, *SOX2-OT*, *PEG3-AS1* *, *LINC00643*			
16 Gr3 vs. 5 normal cerebellum tissue		NA; 149 DE			
	upregulated	*OTX2-AS1*, *BLACAT1*, *LINC00348*, *LINC01355*, *DLEU2*			
	downregulated	*XIST*, *lnc-DLK1*, *SOX2-OT*, *LINC00844*, *MIR100HG*			
39 Gr4 vs. 5 normal cerebellum tissue		NA; 150 DE			
	upregulated	*LINC01419*, *OTX2-AS1*, *BLACAT1*, *DLEU2*, *LINC01355*			
	downregulated	*XIST*, *SOX2-OT*, *MALAT1*, *LINC00643*, *LINC00844*			
58 SHH vs. 164 non-SHH patients **		7829 genes; 328 DE	Publicly available dataset (http://r2.amc.nl)	NA	[25]
	NA ***	*OTX2-AS1*, *HHIP-AS1*, *BLACAT1*, *ENSG00000204789*, *ZFHX4-AS1*			
Gr3 vs. all other MB patients		NA; NA	Publicly available dataset (EGAS00001000215)	NA	[26]
	upregulated	*lnc-STAP1-13*, *lnc-MYO3A-1*, *lnc-SLITRK1-1*, *lnc-AXIN1-1*, *lnc-POU5F1B-5*			
45 Gr3 vs. 66 Gr4 patients		52,128 lncRNAs ****; 1940 DE	Publicly available datasets (EGAD00001003279, MAGIC dataset (Cavalli et al., 2017))	RNA-seq (Illumina TruSeq), RT-qPCR	[27]
	upregulated	*lnc-FOXA1-4*, *lnc-CNR1-1*, *lnc-STAP1-13*, *lnc-ADCY2-10*, *lnc-EIF1-4*			
	downregulated	*lnc-FAM49A-14*, *lnc-PCGF2-3*, *lnc-CISD3-1*, *lnc-VSNL1-5*, *lnc-BAG6-1*			[28]

DE: differentially expressed; NA: information not available; *: HGNC symbol; **: non-SHH patients = Gr3 and Gr4 patients; ***: fold-change information is not available. The comparison between SHH and non-SHH patients was performed based on the correlation coefficient between the expression in SHH vs. non-SHH patients; ****: number of lncRNAs quantified in MB patients from all four subgroups. Due to the extensiveness of the results, Table 1 shows the top five upregulated and downregulated lncRNAs from each study, except for the study of Mutlu et al. [24].

**Table 2 cancers-15-04686-t002:** Significantly DE circRNAs as biomarkers for the diagnosis of MB that were analyzed in at least two different studies.

DE circRNAs	Comparison	Expression	circRNA Source	Method	Reference
*circSKA3*	15 MB vs. 15 controls	upregulated	Tissue RNA (TRIzol reagent)	qRT-PCR (SYBR Green)	[31]
	4 MB vs. 4 controls	upregulated	Tissue RNA (TRIzol reagent)	qRT-PCR (SYBR premix)	[32]
	20 MB vs. 20 controls	upregulated	Tissue RNA (TriQuick reagent)	qRT-PCR (SYBR premix)	[33]
	37 MB vs. 13 controls	upregulated	Tissue RNA (TRIzol reagent)	qRT-PCR (SYBR Green)	[34]
*circSMARCA5*, *circLPAR1*, *circFAM13B*, *circHIPK3*,	1 MB vs. 3 controls	downregulated	Tissue RNA (TRIzol reagent)	RNA-seq (Illumina NovaSeq 6000)	[35]
	40 MB vs. 11 controls	downregulated	CSF RNA (miRNeasy Mini Kit, Qiagen)	RNA-seq (KAPA)	[36]
	35 MB vs. 12 controls	NA	Tissue RNA (TRIzol reagent)	RNA-seq (Illumina NovaSeq 6000)	[37]
*circASXL1*, *circATXN10*, *circCDYL*, *circZKSCAN1*, *circLRBA*	1 MB vs. 3 controls	downregulated	Tissue RNA (TRIzol reagent)	RNA-seq (Illumina NovaSeq 6000)	[35]
	35 MB vs. 12 controls	NA	Tissue RNA (TRIzol reagent)	RNA-seq (Illumina NovaSeq 6000)	[37]

CSF: cerebrospinal fluid; DE: differentially expressed; upregulated: significantly increased expression in MB vs. control group; downregulated: significantly reduced expression in MB vs. control group; NA: information not available (the original article did not include log2FoldChange data).

**Table 3 cancers-15-04686-t003:** CircRNAs as biomarkers for subgroup classification in MB in Azatyan, A. et al., [38].

Comparison	Expression	circRNA
6 WNT vs. 5 controls	upregulated	*circATP8A1-1/2*, *circEPAH3*, *circPRELID2*, *circRMST-2*
	downregulated	*circADGRBR-1/2*, *circSYNEI1*, *circSNHG14*, *circRIMS1*
23 SHH vs. 5 controls	upregulated	*circPCNT*, *circZEB1*, *circBRIP1*, *circSATB2*, *circADGRA3*
	downregulated	*circMAP7*, *circERC1*, *circZHX3*, *circCDYL2*, *circSYNEI1*, *circMCU*, *circPHF21A*
17 Gr3 vs. 5 controls	upregulated	*circANO2*, *circEYS*, *circKHDRBS2*, *circCACNA2D1-1/2*
	downregulated	*circPHF21A*, *circMYH10*, *circRIMS1*, *circMCU*, *circCDYL2*, *circSINEY1*
35 Gr4 vs. 5 controls	upregulated	*circGRM8*, *circDGKB*, *circKHDRBS2*, *circSLCO5A-1/2*
	downregulated	*circARHGAP26*, *circPHF21A*, *circMCU*, *circCDYL2*, *circMTCL1*

Upregulated: significantly increased expression in subgroup vs. control group; downregulated: significantly reduced expression in subgroup vs. control group. RNA was extracted from cerebellum samples for RNA sequencing (Illumina NovaSeq 6000 S4 platform).

**Table 4 cancers-15-04686-t004:** Role of *lnc-IRX3-80* and *lnc-LRRC47-78* in MB tumorigenesis.

LncRNA	Cell Line	Method	Proliferation	Apoptosis	Colony Formation	Invasion	Migration	Cell Cycle Arrest	miRNA Target	Quality Rating	Reference
*lnc-IRX3-80*	DAOY, D341	*lnc-IRX3-80* knockdown	−	+	−	NA	NA	+	NA	Q3	[48]
	DAOY, D341 *	*lnc-IRX3-80* knockdown	−	+	−	−	−	NA	*miR-29c-3p* downregulation	Q3	[49]
*lnc-LRRC47-78*	DAOY	*lnc-LRRC47-78* knockdown	−	+	−	NA	−	NA	NA	Q3	[30]
	DAOY, D341	*lnc-LRRC47-78* knockdown	−	+	NA	−	−	NA	*miR-494-3p* downregulation	Q1	[29]

−: reduced proliferation/apoptosis/invasion/migration/cell cycle arrest rates when the lncRNA is knocked down; +: increased proliferation/apoptosis/invasion/migration/cell cycle arrest rates when the lncRNA is silenced; NA: information not available; * cell lines exposed to cisplatin.

**Table 5 cancers-15-04686-t005:** Role of *circSKA3* in MB cell tumorigenesis.

Cell line	Method	Proliferation	Apoptosis	Invasion	Migration	Cell Cycle Arrest	miRNA Target	Quality Rating	Reference
DAOY, ONS-76	*circSKA3* knockdown	−	+	−	−	+	*miR-383-5p* downregulation	Q1	[33]
DAOY	*circSKA3* knockdown	−	NA	−	−	+	*miR-520* downregulation	Q2	[31]
	*circSKA3* overexpression	+	NA	+	+	−			
DAOY, D283	*circSKA3* knockdown	−	+	−	−	+	*miR-326* downregulation	Q2	[34]
	*circSKA3* overexpression	+	−	+	+	−			
DAOY	*circSKA3* knockdown	−	NA	−	−	NA	NA	Q3	[32]

−: reduced proliferation/apoptosis/invasion/migration/cell cycle arrest rates when *circSKA3* is overexpressed/knocked down; +: increased proliferation/apoptosis/invasion/migration/cell cycle arrest rates when *circSKA3* is overexpressed/silenced; NA: information not available.

## Data Availability

The data presented in this study are available in this article (and Appendix A).

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
