# Peer review of "The Role of the Dysregulation of Long Non-Coding and Circular RNA Expression in Medulloblastoma: A Systematic Review"

_cancers, 2023, doi:10.3390/cancers15194686_

Round 1

Reviewer 1 Report

The "The role of the dysregulation of long non-coding and circular 2 RNAs expression in medulloblastoma: A systematic review" is well designed and in the scope of this journal. 

As such review article is well in writing, but I suggest adding the diagram to present the mechanism or expression of RNAs in medulloblastoma in the introduction section. 

There are a few English spelling and grammar mistakes, please read it carefully before resubmission. 

Minor English language issues. few English spelling and grammar mistakes are there. 

Author Response

We thank the reviewer for the suggestion to include a figure with the general mechanisms of action of lncRNAs and circRNAs. Therefore we added the Figure 1 to the manuscript.

Regarding the mentioned English spelling and grammar mistakes, an English writing assistant was used.

Reviewer 2 Report

Overall, this is a compelling review paper that depicts the different expression lncRNAs and circRNA, and introduces some lncRNAs and circRNAs as biomarkers and have some functions in regulating genes expression especially as miRNA sponge,which is shown to contribute to the medulloblastoma is a surprising way. I have one suggestion for "3.3. Functional studies in medulloblastoma", I think it would be better show more examples of how these noncoding RNA regulate genes expression and the mechanism actions.Noncoding RNA has multiple functions not only as miRNA sponge,they can work through RNA binding protein and others,and the function of these ncRNA depend on their subcellular locallization. That may provide more informations and more comprehensive for the audience. 

Author Response

We thank the reviewer their suggestion. In the Introduction section, we have added Figure 1 and further explained the potential roles of lncRNAs and circRNAs. However, none of the papers reviewed studied this function for lncRNAs and circRNAs. We consider it very interesting, and when possible we have suggested functions related to RBPs, especially for circRNAs. Due to the lack of data analyzing other functions apart form miRNA sponge, we included this aspect in conclusion and further studies.

Reviewer 3 Report

The topic of the study is relevant, as non-coding RNAs are an emerging field in cancer research. The authors tried to systematically identify and review published studies on lncRNAs and circRNAs in pediatric brain cancer. Overall, this study aims to determine the potential of these non-coding biomarkers in diagnosis, classification and prognosis of pediatric brain cancer. Despite some strengths, improvements are required regarding methodology, data analysis and reporting, study limitations.

1.The authors of this systematic review searched two databases to identify relevant articles: PubMed and Scopus. Using a limited number of databases increases the risk of publication and search bias, as different databases cover different journals and publications. Researchers should aim to utilize multiple reputable and relevant databases to maximize the chances of finding all pertinent studies. Adding databases like Web of Science and Cochrane can make the search more comprehensive by covering more journals and publications. Not using key databases may result in missing relevant studies indexed in those databases, threatening the validity and generalizability of the findings.

2. In the discussion and conclusion sections, a logical connection between the different findings is not well established and a comprehensive interpretation of the results is not provided. More specifically, the associations between the various deregulated lncRNAs and circRNAs identified across the subgroups are not clearly delineated. The implications of the overlapping and distinct expression profiles on the underlying biology and pathogenesis of each subtype are not sufficiently discussed. Furthermore, the functional importance of the top deregulated candidates is not adequately interpreted in light of the discovery study results. The discussion does not provide an integrated picture of how the expression patterns may relate to the functional mechanistic studies. Establishing logical connections between the different components of evidence is essential for a comprehensive and insightful discussion of the systematic review findings.

3. Overall, the English language quality is fair but needs improvement in certain areas. On the positive side, the language is generally clear and comprehensible. Basic grammar and vocabulary usage are adequate for a scientific publication. The language flows reasonably well in most sections like the introduction and methods. However, there are some problematic areas that should be addressed: The abstract needs to be revised for clearer and more concise writing. There is verbosity and redundancy in places. The results section lacks clarity in presenting key findings from the different studies. Better organization and explanatory details are required. The discussion writing is quite weak. Logical connections are missing, and clinical/research implications are not sufficiently elaborated. There are stylistic issues like long, convoluted sentences and overuse of passive voice in some paragraphs. Shorter, active voice sentences will improve readability.

Overall, attention needs to be given to improve the clarity, conciseness and flow of the writing, especially in the abstract, results and discussion sections. Refining the clinical/research interpretations and inclusion of explanations and definitions will also enhance the quality. With systematic editing and revisions, the language quality can be improved to meet publication standards.

Author Response

We thank the reviewer for the impressions. We will answer all the comments one-by-one below.

  1. We understand the concern of the reviewer regarding the possibility of having missed studies which would meet the inclusion criteria. Following the reviewer’s suggestion, we entered our search terms in Web of Science and Cochrane databases. Concerning lncRNAs, , we found 43 records in Web of Science and 0 records in Cochrane database. With respect to circRNAs, we found 19 records in Web of Science and 0 records in Crochrane databases. Because these searched failed to add new articles to our study, we decided to maintain the original design.

2. The discussion of lncRNAs has been edited and updated accordingly. More explanations and associations were given for the top deregulated candidates. Nevertheless, with respect to circRNAs, when referring to the comparisons among subgroups, only an article compared the expression between different subtypes and controls. Therefore, we decided to focus on the results on this study. 

3. Following the suggestion of the reviewer, we have revised the writing of the abstract, results, and discussion. Elaborations on the clinical/research implications were added throughout the discussion. 

Reviewer 4 Report

 “The role of the dysregulation of long non-coding and circular RNAs expression in medulloblastoma: A systematic review” is a much needed review of the long non-coding RNA and circRNA literature of medulloblastoma.  There are only minor revisions:

1-The text and figure 1 do not match up. Based on the numbers from the figure, assuming that 2 papers are in both columns, that should leave 18 papers for lncRNA, not 19 as is stated in the text. One of the numbers is incorrect.

2-Table 1 is difficult to read. I would suggest putting dividers between each study or repeating the lncRNA source, method and reference for each entry.

3-Table 2 is difficult to read, please separate the circRNAs and their respective experiments in a clearer fashion.

4-Table 4 is difficult to read. I would suggest having the column headings in a vertical position to make space for the cell line information ("DAOY" should not be split on two lines), miRNA target and method.

5-The authors need to properly introduce abbreviations such as “DE” and “ceRNA”.

6-Line 376, HHIP-AS1 is misspelled.

7-In the abstract, "noncoding" is not consistent with the spelling of non-conding throughout the text.

8-The authors need to be careful as there are commonly only four consensus subgroups of MB (WNT, SHH, Group 3 and Group 4) or four WHO subgroups (WNT, SHH-TP53 wild type, SHH-TP53 mutant and non-WNT, non-SHH medulloblastoma). Depending on the study, there could be 11-12 subtypes of medulloblastoma and I do not think the authors are referring to these. Therefore, I would suggest using the term “subgroup” instead of “subtype” in this manuscript.

9-Since there is so much effort spent on classifying medulloblastoma into their respective subgroups, I am surprised that this has not been performed to date using circRNAs or, if it has happened, then why are these data not reported in this systematic review?

Although an interesting article, the English needs to be updated as it makes the article difficult to read. For instance, there are missing commas, specifically after “and” such as line 48 where it should be written “... and, more recently, multiomic studies ...” where “more recently” is a separate clause. There are many such examples throughout the text. Additionally, there are certain words which should be changed such as stablished (established), warred (warranted) and carcinogenic (oncogenic).

Author Response

First, following the suggestion of the reviewer, we have revised and corrected the English style.

These will be answered below.

1. We thank the reviewer for the correction. We have modified the text, 18 articles was the correct number:

“Finally, 18 articles were included in the review, of which three performed lncRNA expression discovery analysis in MB patients, 13 performed functional analysis of lncRNAs, and two performed both discovery and functional study.”  

2. Following the reviewer’s suggestion, we added dividers to table 1 to improve its readability.

3. We have separated each of the circRNAs studied and each of the experiments for each of the circRNA with dividers to increase readability.

4. We have adjusted the table with vertical column headings and more space for the rest of the information.

5. 

Such abbreviations have been checked and properly corrected:

DE: differentially expressed

“MIR100HG was further analyzed in a functional study, in which it was shown to de-repress the expression of the major driver genes in Gr4 MB, CDK6, MYCN, SNCAIP, and KDM6A, by acting as competing endogenous RNAs (ceRNA) and binding to three miRNAs: miR-19a-3p, miR-19b-3p, and mir-106a-5p.”

6. We checked and corrected this misspelling error.

7. We checked and corrected the spelling of “noncoding” to “non-coding” in the abstract.

8. We checked and modified the term subtype into subgroup throughout the whole manuscript.

9. We only found an article comparing the expression between different subgroups and controls. Therefore, circRNAs differentially expressed in MB subgroups are still unclear and more studies are required to reach a robust conclusion on this issue. We highlight this issue in conclusion and future studies section.

Reviewer 5 Report

The systematic review by Ivan et.al. provides an overview of the current research on lncRNA and circRNA in the field of medublloblastoma. The authors provide method and criteria for selection of articles related to lncRNA and circRNA in medulloblastoma. The authors provide a classification of articles based on established metric as well as inhouse metric. The authors report previous studies documenting diagnostic potential of lncRNA and circRNA followed by discussion on the functional studies done previously to understand mechanistic role of these RNA species in medulloblastoma. Finally authors provide speculative functional mechanisms of the lncRNA and circRNA driven regulation in medulloblastoma. Following are the merits and demerits of the article. The article has comprehensively reviewed articles published documenting the diagnostic as well as functional role of lncRNA and circRNA in medulloblastoma.

However, there are some major concerns that need to be addressed.

1.       Given that the review focuses on the functional mechanism of the lncRNA and circRNA, authors fail to provide details on the general mechanisms of actions of lncRNA and circRNA in various molecular functions like gene regulation, transport, sponging, RBP and recruitment, etc. Authors can provide details on the mechanism of action and cite appropriate reviews detailing the functions.

2.       The authors present a novel metric for establishing the quality of the paper. However, the purpose of this metric is unclear. Is this used for inclusion or exclusion of the studies from the review?

3.       The metric is confusing and does not justify the values assigned to the criteria. E.g., why “the analysis of at least two cancer phenotypes” is 1pt while “the inclusion of rescues experiments” has 2 pts. The metric does not provide information on which criteria were met in the study, nor does it provide relative measure of the quality of paper as evidenced by diagnostic studies in lncRNA where all the articles are of same measure. Studies with larger cohort, age matched controls and independent validation clearly have higher quality.

4.       The authors speculate functional mechanisms of RNA species based on glioma and hepatocellular carcinoma without justification for the comparison or relative measures of expression of effector molecules dues to dysregulation of the lncRNA/circRNA. It would be helpful to also check whether the levels of effectors are similar or different in different cancers (e.g. miR-17-3p/miR-181-5p and miR-4295).

5.       Authors should include that the comparisons are hypothetical and need investigation where mentioned. E.g. page 12 line 408, page 13 line 462, page 13 line 478, etc.

6.       There is redundancy in the manuscript where the authors discuss same things multiple times. E.g. page 11 line 374 and page 11 line 514. Consider a rewrite.

Minor

It is unclear what the authors want to convey in page 2 line 47. Are the authours suggesting that histopathological risk classification leads to improved overall survival? How?

Page 2 line 53 please use full forms for group 3 (Gr3) and group 4 (Gr4).

Use consistent naming throughout the paper. E.g. page 11 line 384 and line 387 (NEAT1 and lnc-NEAT1).

Minor revision required.

Author Response

We thank the reviewers all the comments and we will go through them one-by-one below:

Major comments responses:

1. In order to give a better explanation of the putative functional mechanisms of lncRNAs and circRNAs, we have included Figure 1 and additional information in the Introduction to explain the general functional mechanisms of lncRNA and circRNAs.

2. Our in-house quality metric aimed to scheme the quality of the evidence found with respect to functional studies and to only include those studies with a minimum quality in their design. Despite grading these studies, we did not have to exclude any article based on this scale. We added the following explanatory comment in the Material and Methods section:

The inclusion criterion was limited to articles falling within the Q1, Q2, or Q3 categories. By encompassing various aspects such as cell line diversity, clinical relevance, and mechanistic insight, we aimed to picture the quality of the functional evidence on lncRNAs and circRNA studies.”

3. Regarding the discovery studies, a previously stablished classification was used as reference. Regarding the functional studies, we wre not able to find any valid classification and we created an in house classification. Each criteria was assigned a different amount of points to reflect its relative relevance in assessing the comprehensive nature of functional analysis studies. We have included this statement in methods.

4. The levels of the different effectors and proposed miRNA targets have not been studied in MB but their expression in other tumors has been checked an added to the manuscripts in different sentences:

“In fact, different studies have already shown that miR-17-3p and miR-181b-5p are usually upregulated in different cancer types such colon and breast cancer respectively [71, 72]. In this context, the downregulation of circSMARCA5 in MB samples could hypothetically explain MB tumor progression, as lower expression of this circRNA would result in higher expression of the mentioned miRNAs. ”

“Because this circRNA is downregulated in MB, if the same mechanism applies, miR-762 would be overexpressed and its target gene IRF7 would be inhibited, as in a study published in breast cancer [76]. Furthermore, another study involving gastric cancer, revealed that this miRNA was aberrantly upregulated in tumor tissue and cell lines. ”

“Additionally, these miRNAs seem to be downregulated in different cancers, such colorectal cancer, confirming the hypothesis that circHIPK3 upregulation may reduce these miRNA levels [84].”

“In this context, a study delved into the functional aspects of NEAT1 in MB and found its role as sponge for miR-23a-3p [38], a miRNA recognized as tumor suppressor frequently downregulated in various cancers [50-52]. Moreover, the NEAT1/miR-23a-3p sponging is validated in multiple cancers, reinforcing the oncogenic significance of this interaction [53-56]. The sponging of miR-23a-3p by NEAT1 in MB leads to upregulation of glutaminase (GLS) which results in increased glutamine metabolism [38].”

5.  The different comparisons and mechanism of actions proposed were referred to as hypothetical. 

6. We thank the reviewer for this comment as it will help to clarify the content of this review. In the discussion, several paragraphs were re-written with this purpose such as:

      “Concerning NEAT1, this lncRNA exhibits overexpression in MB patients compared to normal tissue [12, 24] and specifically within the SHH or Gr3 subgroups [24, 12]. The overexpression of NEAT1 plays a crucial oncogenic role in other solid tumors, such as lung cancer and breast cancer [49]. This suggests that NEAT1 also might function as a significant oncogene within MB. In this context, a study delved into the functional aspects of NEAT1 in MB and found its role as sponge for miR-23a-3p [38], a miRNA recognized as tumor suppressor frequently downregulated in various cancers [50-52]. Moreover, the NEAT1/miR-23a-3p sponging is validated in multiple cancers, reinforcing the oncogenic significance of this interaction [53-56]. The sponging of miR-23a-3p by NEAT1 in MB leads to upregulation of glutaminase (GLS) which results in increased glutamine metabolism [38]. Interestingly, this metabolic shift is correlated with cisplatin resistance in MB cells [38]. This association implies that this lncRNA might contribute to an unfavorable treatment response, aligning with observations in other tumor types [49]. NEAT1`s oncogenic influence goes beyond miRNA interactions. NEAT1 forms structural domains that enable it to interact with RBPs, contributing to the formation paraspeckles. These interactions further contribute to its multifaceted role in gene regulation, nuclear organization, and cancer progression. These paraspeckles impact cancer by sequestering RBPs like SPFQ, suppressing apoptosis in certain contexts and promoting chemoresistance, especially under hypoxic conditions commonly found in solid tumors [57]. However, the specific interaction between NEAT1 and RBPs in the context of MB remains an area that warrants further research and investigation.”

Lnc-LRRC47-78 (also known as TP73-AS1) was upregulated in MB patients when compared to healthy individuals [36, 48]. Additionally, lnc-LRRC47-78 exhibited DE in SHH versus non-SHH patients [25]. Within this subgroup, in specific TP53 wild-type subgroup, high expression of lnc-LRRC47-78 was correlated with poorer prognosis, highlighting its potential as prognostic indicator [48]. Interestingly, in a particular subgroup of glioblastomas associated with a more favorable prognosis, the lnc-LRRC47-78 promoter is hypermethylated, resulting in its downregulation [59]. In MB cells, the lnc-LRRC47-78 gene is hypomethylated leading to its upregulation which elevates the proliferation, migration, and survival rates [48]. This indicates that changes in methylation are key influences on the activity of lnc-LRRC47-78 in brain tumors, including MB. These changes, whether causing upregulation or downregulation, are most probably closely associated with patients’ outcomes, highlighting the potential of lnc-LRRC47-78's methylation status as a critical indicator for both prognosis and potential therapeutic avenues. According to the functional studies performed in MB, lnc-LRRC47-78 serves as an oncogene via sponging miR-494-3p, thus elevating the expression of EIF5A2 [36]. Interestingly, the activity changes in EIF5A have been linked to cancer development, particularly its higher expression in GBM patient samples compared to normal glia cells [60]. However, further research is warranted to better understand the mechanisms involved in the lnc-LRRC47-78/miR-494-3p/EIF5A2 axis in MB for potential targeted therapeutic approaches. “

“These miRNAs regulate the expression of PTEN, which can negatively regulate PI3K/AKT signal transduction, thus inhibiting or promoting tumor progression [70]. In fact, different studies have already shown that miR-17-3p and miR-181b-5p are usually upregulated in different cancer types such colon and breast cancer respectively [71, 72]. In this context, the downregulation of circSMARCA5 in MB samples could hypothetically explain MB tumor progression, as lower expression of this circRNA would result in higher expression of the mentioned miRNAs. As a result, PTEN expression would be reduced, resulting in a constitutively active PI3K/AKT signaling pathway and potential increase in tumor proliferation and invasion.”

“Similarly, to circSMARCA5, circLPAR1 may contribute to its role through miRNA targeting as it has been shown to target miR-762, which is a known oncomiR, in bladder cancer [74, 76]. Because this circRNA is downregulated in MB, if the same mechanism applies, miR-762 would be overexpressed and its target gene IRF7 would be inhibited, as in a study published in breast cancer [76]. Furthermore, another study involving gastric cancer, revealed that this miRNA was aberrantly upregulated in tumor tissue and cell lines. This deregulation increases PI3K/AKT signaling pathway activity, and, thus, promotes cancer cell proliferation [77]. In summary, the deregulation of this gene and potential miRNA targets, could be a plausible mechanism of action concerning the downregulation of this circRNA in MB. However, further studies regarding this miRNA and other potential mechanisms like interactions with RBPs are needed in MB samples.”

Minor comments responses:

1. We thank the reviewer for this comment. In that, quote we wanted to highlight the relevance of the improvement in the MB subgroup classification, as each subtype has different genetic features, which could directly affect patient treatment or survival. We understand that the histopatologic features have not considered for risk stratification straightforward as the genetic and cytogenetic aberrations (Figure 4, Juraschka and Taylor 2019). These genetic features have enabled the four-subgroup classification, which have started to be informative for treatment election. Indeed, Low-risk WNT patients receive reduced-intensity therapy, skeletally mature SHH patients receive vismodegib (in addition to standard of care treatment), and standard- risk and high-risk non-WNT/non-SHH patients are prioritized for treatment intensification with pemetrexed and gemcitabine. Therefore, we will re-write the sentence.

“Multi-omic studies, including whole exome sequencing (WES), transcriptomics (RNA-seq), and methylation, allowed the classification of MB into four different molecular subgroups which are being considered for risk classification which led to the improvement of the overall survival of MB patients due to different treatment approaches. “

2. These suggestions were checked and modified trough the paper. 

Round 2

Reviewer 5 Report

NA